# Macrophage-like Blood Cells Are Involved in Inter-Tissue Communication to Activate JAK/STAT Signaling, Inducing Antitumor Turandot Proteins in *Drosophila* Fat Body via the TNF-JNK Pathway

**DOI:** 10.3390/ijms252313110

**Published:** 2024-12-06

**Authors:** Juri Kinoshita, Yuriko Kinoshita, Tadashi Nomura, Yoshihiro H. Inoue

**Affiliations:** 1Biomedical Research Center, Kyoto Institute of Technology, Matsugasaki, Sakyo, Kyoto 606-0962, Japan; m3641010@edu.kit.ac.jp (J.K.); m2641013@edu.kit.jp (Y.K.); tadnom@kit.ac.jp (T.N.); 2Graduate School of Science and Technology, Kyoto Institute of Technology, Matsugasaki, Sakyo, Kyoto 606-8585, Japan

**Keywords:** *Drosophila*, hematopoietic cell tumor, innate immune system, cytokines, JNK pathway, TNF family, JAK-STAT pathway

## Abstract

**Abstract:** Turandot (Tot) family proteins, which are induced via the JAK/STAT pathway after infection, also suppress lymph gland tumors in *Drosophila mxc^mbn1^* mutant larvae. We investigated the potential role of hemocytes in *Tot* induction in tumor-bearing mutants via immunostaining and RNAi experiments. Normal hemocytes transplanted into mutant larvae were recruited to the tumor and fat body (FB), suggesting that these cells transmit tumor-related information. The transplanted hemocytes ectopically expressed Unpaired3 (Upd3), which is necessary for the activation of JAK/STAT. Eiger, a *Drosophila* tumor necrosis factor (TNF) ortholog, was highly expressed in tumors. Depletion of the Eiger receptor in hemocytes reduced *Tot* levels and eventually enhanced tumor growth. The c-Jun N-terminal kinase (JNK) pathway, acting downstream of the receptor, was also activated in the hemocytes of mutants. Downregulation of the JNK pathway in hemocytes inhibited *Tot* induction, leading to enhanced tumor growth. These results suggest that *upd3* expression in hemocytes depends on the Eiger–JNK pathway. We propose that after Eiger activates the JNK pathway in hemocytes present on the tumor, cells expressing Upd3 are recruited to the FB. Upd3 then activates JAK/STAT to induce the expression of antitumor proteins. This study highlights the intricate communication between tissues via blood cells during tumor suppression.

## 1. Introduction

Innate immunity is the first line of defense against infectious pathogens and tumor cells. Although insects, including *Drosophila*, do not have an acquired immune system, they exhibit immunological responses against microbes, parasite eggs, wounds, and tumors [1]. When microbial pathogens enter the body, humoral factors are induced. The fat body (FB) produces secreted proteins of a low molecular weight, such as antimicrobial peptides (AMPs) and Turandot proteins (Tots) [2,3,4]. Depending on the type of invader (fungi, Gram-negative bacteria, or Gram-positive bacteria), different proteins are produced in the FB. Each AMP is induced by one or both Toll- and Imd-mediated signaling pathways [4], homologous to the mammalian Toll-like receptor (TLR) and tumor necrosis factor (TNF) pathways, respectively [5,6]. Five major AMPs (Drosomycin, Defensin, Diptericin, Metchnikowin, and AttacinA) are induced in tumor-bearing mutant larvae and have cytotoxic effects against tumors [7,8,9].

In addition to these well-known innate immunity pathways, the JAK/STAT signaling pathway plays a critical role in inducing another immune-responsible protein family called the Tot family after infection [10]. When Unpaired (Upd) cytokines such as Upd3 bind to their receptor (Domeless (Dome)), the JAK/STAT pathway is activated in the cytoplasm [11]. Consequently, the transcription of *Tot* family genes is induced in the FB [6,12]. Similarly to AMPs, TotB and TotF proteins induced by the JAK/STAT pathway also have cytotoxic effects against tumors. These Tot proteins are induced in the FB and incorporated into circulating hemocytes [13]. As a result, apoptosis is observed in the tumors generated in the lymph gland (LG) (LG tumors) due to the induction of AMPs and Tots, and tumor cell proliferation is also inhibited in Tots. However, the mechanism by which Tots affect LG tumors located away from the FB has not yet been characterized.

The c-Jun N-terminal kinase (JNK) pathway is also activated in response to diverse intracellular stresses, including tumors [14]. When the cytokine called Eiger, corresponding to a mammalian TNF family protein, binds to its receptors composed of Wengen and Grindelwald, this activates the JNK signaling pathway in the cytoplasm, including Hep, corresponding to JNKK, and Bsk, corresponding to JNK [15]. Activated Bsk phosphorylates and activates the transcription factor Jra, which induces the transcription of *upd* genes encoding ligands for the JAK/STAT pathway [16]. However, the interaction between these signaling pathways in *mxc^mbn1^* larvae remains to be elucidated.

Hemocytes in the hemolymph, which are classified as plasmatocytes, lamellocytes, and crystal cells, also play critical roles in eliminating pathogens [17]. Plasmatocytes, the equivalent of mammalian macrophages, comprise 95% of all *Drosophila* hemocytes and play essential roles in cellular immune responses such as phagocytosis [18,19]. During the larval stage, new hemocytes are produced in a hematopoietic tissue called the LG [20]. In larvae hemizygous for a loss-of-function mutation in the *multi sex combs* (*mxc*) gene, the hemocyte precursor cells in the LG over-proliferate and differentiate abnormally [7,21,22,23]. The hemocyte precursor cells in the LG of *mxc^mbn1^* mutant larvae become malignant and invade other tissues [21,24]. When mutant LG cells are transplanted into the abdominal cavity of normal flies, they continue to proliferate and infiltrate other adult tissues. Some aberrant hemocytes expressing undifferentiated markers are released into the hemolymph. These leukemia-like phenotypes are lethal to mutant larvae from the third instar larval stage to the pupal stage [7,23,24]. In response to LG tumors, three innate immune pathways, the Toll-mediated, Imd-mediated, and JAK/STAT pathways, are activated to induce five major AMPs and four Tots in the FB of *mxc^mbn1^* larvae [7,13]. Consequently, these AMPs stimulate apoptosis specifically in LG tumors [7,8,9]. For AMPs to be induced in the FB far away from the LG tumor, the information relevant to the presence of tumor cells must be transmitted to the FB. Mutant hemocytes, as well as normal hemocytes transplanted from control larvae, are preferentially associated with LG tumors in mutant larvae [9]. However, it has not been investigated whether hemocytes are efficiently recruited to the FB in *mxc^mbn1^*. The possibility that the recruitment of hemocytes to this tissue is due to the tumor characteristics of the mutant hemocytes has not been ruled out. To clarify, we investigated whether hemocytes transplanted from normal larvae were recruited more efficiently to mutant FBs than to control tissue.

TotB and TotF depletion compromises apoptosis and enhances tumor cell proliferation [13]. However, the mechanism by which the JAK/STAT pathway is activated in FBs has not been studied. Therefore, we aimed to clarify whether circulating hemocytes play a critical role in conveying tumor information to the FB to activate the JAK/STAT pathway. Upd3, a JAK/STAT pathway ligand produced by tumor cells, signals the immunocompetent tissue to activate the JAK/STAT pathway [11]. Our previous study also showed that hemocytes from *mxc^mbn1^* larvae ectopically express Upd3; consequently, this pathway is activated [13]. Thus, we also aimed to elucidate the mechanism by which the JAK/STAT signaling pathway is activated in FBs in response to LG tumors via circulating hemocytes.

In the present study, we first examined whether *upd3* expression was required to induce Tot mRNA expression in FB, that is, the activation of the JAK/STAT pathway. We further investigated the effects of ectopic expression on LG tumor growth. Second, we focused on the mechanism by which Upd3 is highly expressed in the hemocytes of *mxc^mbn1^* larvae. The TNF-like cytokine Eiger is expressed in the tumors of *Drosophila* imaginal discs and activates the JNK pathway [25]. Therefore, we investigated whether Eiger was highly expressed in the LG tumors of *mxc^mbn1^* larvae. Third, we examined whether Eiger expression in LG tumors was required to activate the JNK pathway in circulating hemocytes. We then assessed whether the JNK pathway downstream of the Eiger receptors in circulating hemocytes was required to activate the JAK/STAT pathway in the FB. Additionally, we examined their effects on the growth of LG tumors. Based on these results, we proposed a model to explain how the higher Eiger expression in LG tumors of *mxc^mbn1^* larvae leads to activation of the JAK/STAT pathway and, ultimately, induction of *Tot* expression in FBs. AMPs and Tots secreted from an FB are taken up by plasmatocytes of mutant larvae [7,13]. Thus, we also determined whether Tot proteins were taken up by macrophage-like plasmatocytes and transported to the tumor. The current results will be important for the future analysis of this tumor suppression mechanism—which may involve inter-tissue communication via hematopoietic cells—in *Drosophila* and other animal models.

## 2. Results

### 2.1. An Increase in Normal Hemocytes Is Associated with the FB in mxc^mbn1^ Mutant Larvae Harboring the LG Tumor

Normal circulating hemocytes are recruited to the LG tumor more efficiently in mutant larvae than in normal larvae when the cells are transplanted into mutant larvae [9]. To understand the mechanism by which information regarding a tumor is transmitted to the FB, we first examined whether more circulating hemocytes were associated with the FB in mutant larvae. We scored the circulating hemocytes labeled with GFP on the FB in mutant *(mxc^mbn1^*/*Y*; *He* > *GFP*) (Figure 1b) and control (*w*/*Y*; *He* > *GFP*) (Figure 1a) larvae and converted the average cell number in each larva to that in a 1 mm^2^ FB area. In total, 61.9 hemocytes (*n* = 25) were observed on average in 1 mm^2^ of the FB area in *mxc^mbn1^* larvae, whereas 1.1 hemocytes (*n* = 22) were observed in the same FB area in normal larvae (Figure 1a,b). The differences were significant (Figure 1c); 60 times more hemocytes were localized on the FB in mutant larvae bearing LG tumors than in control larvae. As 4 times more hemocytes were contained in the hemolymph of the mutant (7.7 ± 1.1 × 10^4^ cells in 1 mL of hemolymph, *n* = 16) than of the control (1.8 ± 0.1 × 10^4^ cells in 1 mL of hemolymph, *n* = 12) larvae, mutant hemocytes were recruited more efficiently to the FB.

To clarify whether normal circulating hemocytes are also recruited to the FB in mutant larvae, we transplanted larval hemolymphs containing GFP-labeled normal hemocytes (*w*/*Y*; *He* > *GFP*) into control and mutant larvae at the third instar stage. Fifteen hours after hemolymph transplantation, in which 1.2 × 10^4^ circulating hemocytes (*w*/*Y*; *He* > *GFP*) were contained on average (*n* = 21) (Figure 1d,e), we counted the number of transplanted GFP+ hemocytes on the FB and converted it to the number of cells per unit area (1 mm^2^) of the tissue. We counted 4.0 hemocytes on average (*n* = 31) per FB area in *mxc^mbn1^* larvae (Figure 1e,f), whereas 1.5 hemocytes (*n* = 31) were in the same area in normal larvae (Figure 1d,f). The increase in mutant larvae was significant (Figure 1f). Confocal microscopy observations confirmed that the transplanted normal hemocytes were associated with the surface (but not the inside) of the FB in normal and mutant larvae (see the images, which are only for review in a repository).

### 2.2. Ectopic upd3 Expression in Normal Hemocytes Transplanted into mxc^mbn1^ Larvae and Hemocyte-Specific Depletion of upd3 Resulted in Reduced TotF mRNA Levels and Increased LG Tumor Growth

To confirm the possibility that hemocytes activate the JAK/STAT signaling pathway in the FB via Upd3, we initially observed ectopic *upd3* expression in normal hemocytes transplanted into *mxc^mbn1^* larvae. We transplanted normal hemocytes harboring the *upd3-LacZ* reporter that monitors gene expression (*w*/*Y*; *He* > *GFP*/*upd3-LacZ*) into control and *mxc^mbn1^* larvae at the third instar stage. Fifteen hours after transplantation, we observed a few anti-β-gal immunostaining signals above the background level (*n* = 73 cells examined) (Figure 2a,a′′′). In contrast, we observed a robust immunostaining signal in GFP^+^ normal hemocytes in *mxc^mbn1^* larvae (*n* = 108 cells examined) (Figure 2b,b′′′). The difference between control and mutant larvae was significant (Figure 2c).

We investigated whether depletion of *upd3* affected *TotF* expression in the FB of *mxc^mbn1^* larvae (Figure 3). We quantified *TotF* mRNA levels via qRT-PCR using RNAs from normal control larvae (*w*/*Y*), *mxc^mbn1^* larvae expressing hemocyte-specific dsRNA against *GFP* mRNA (*mxc^mbn1^*/*Y*; *He* > *GFPRNAi*) as a control for depletion, and *mxc^mbn1^* larvae expressing hemocyte-specific *upd3* depletion (*mxc^mbn1^*/*Y*; *He* > *upd3RNAi*). The *TotF* level in *mxc^mbn1^* larvae with hemocyte-specific *upd3* depletion was reduced to 55.9% of that in *mxc^mbn1^* larvae without the depletion. This difference was significant (Figure 3a).

We further investigated whether *upd3* depletion in hemocytes influences the hyperplasia of LG tumors in *mxc^mbn1^* larvae. The average LG sizes of the normal control, *mxc^mbn1^* larvae (*mxc^mbn1^*/*Y*; *He* > *GFPRNAi*), and *mxc^mbn1^* larvae harboring the *upd3* depletion (*mxc^mbn1^*/*Y*; *He* > *upd3RNAi*) were 0.04 mm^2^ (*n* = 20), 0.36 mm^2^ (*n* = 20) and 0.53 mm^2^ (*n* = 17), respectively (Figure 3e). The LG size of *mxc^mbn1^* with *upd3* depletion (*mxc^mbn1^*/*Y*; *He* > *upd3RNAi*) was more than 1.4 times higher than that of *mxc^mbn1^* (*mxc^mbn1^*/*Y*; *He* > *GFPRNAi*). This difference was significant (one-way ANOVA with Bonferroni correction).

### 2.3. Hyper-Activation of JNK in Normal Hemocytes Transplanted into mxc^mbn1^ Larvae and Its Role in Inducing TotF Expression and Suppressing Tumor Growth

We next investigated whether the JNK pathway was activated in normal hemocytes transplanted into *mxc^mbn1^* larvae. Fifteen hours after the transplantation of the normal hemocytes (1.2 × 10^4^ cells) (*w*/*Y*; *He* > *GFP*) into normal larvae (*w*/*Y*), we detected only a few immunostaining signals with the anti-phosphorylated JNK antibody over the background level in GFP+ transplanted cells (*n* = 88 cells examined) (Figure 4a′′′). In contrast, we observed a strong immunostaining signal in GFP+ normal hemocytes in the hemolymph of *mxc^mbn1^* larvae (*n* = 96 cells) (Figure 4b′′′). The differences in the signal intensity corresponding to the extent of the JNK activation between the control and mutant larvae were significant (Figure 4c). Moreover, MMP1, another downstream target of the JNK pathway, was also induced in normal hemocytes transplanted into mutant larvae (Appendix A).

As the JNK pathway can induce the expression of Upd3 during development [26], we next investigated whether the induction of *TotF* expression in the FB depended on the JNK pathway in the hemocytes of *mxc^mbn1^* larvae. First, we examined whether the JNK pathway was activated and thus induced *TotF* expression in the FB of normal control larvae (*w*/*Y*) and *mxc^mbn1^* larvae expressing control dsRNA against *GFP* mRNA (*mxc^mbn1^*/*Y*; *He* > *GFPRNAi*), of mutant larvae harboring the hemocyte-specific *hep* depletion (*mxc^mbn1^*/*Y*; *He* > *hepRNAi*) and of larvae harboring *bsk* depletion (*mxc^mbn1^*/*Y*; *He* > *bskRNAi*). The *TotF* mRNA levels in *mxc^mbn1^* with the hemocyte-specific depletion of *hep* and *bsk* were reduced to an average of 39.6% and 24.3% of those in *mxc^mbn1^*, respectively (Figure 5a,b). This difference was statistically significant (Figure 5a,b).

We further investigated whether the depletion of JNK in the hemocytes of *mxc^mbn1^* larvae enhanced the hyperplasia of LG tumors (Figure 5c–g). The average LG size of normal control (*w*/*Y*) and *mxc^mbn1^* (*mxc^mbn1^*/*Y*; *He* > *GFPRNAi*) larvae at the third-instar stage was 0.04 mm^2^ (*n* = 20) and 0.39 mm^2^ (*n* = 20), respectively. The LGs of *mxc^mbn1^* were more than 12 times larger on average than those of the controls (Figure 5g). LG sizes further increased in *mxc^mbn1^* larvae harboring the hemocyte-specific depletion *hep* (*mxc^mbn1^*/*Y*; *He* > *hepRNAi*) and those harboring the depletion of *bsk* (*mxc^mbn1^*/*Y*; *He* > *bskRNAi*) to 0.48 mm^2^ and 0.50 mm^2^, respectively (Figure 5e,f). The differences between these sizes and the LG sizes of *mxc^mbn1^* larvae without these depletions were statistically significant (Figure 5g).

### 2.4. Ectopic Expression of Eiger, a TNF Superfamily Ligand, in LG Tumors of mxc^mbn1^ Mutant Larvae

To understand the mechanism by which the activation of the JNK pathway occurs in hemocytes, which leads to *TotF* induction in the FB, we investigated whether Eiger is involved in JNK activation in *mxc^mbn1^* larvae. We examined Eiger expression in the LG tumors of mutant larvae via immunostaining with an anti-Eiger antibody (Figure 6a,b). The average fluorescence intensity of whole-lobe LG regions was calculated from immunostaining images of LGs from control (*w*/*Y*) and *mxc^mbn1^* (*mxc^mbn1^*/*Y*) larvae at the third instar stage. The average fluorescence intensity was 5 times higher in mutant LGs (*n* = 50) than in the normal control LGs (*n* = 42). This difference was significant (Figure 6c). These results indicate that Eiger expression was considerably higher in LG tumors of *mxc^mbn1^* than in control LGs.

### 2.5. Depletion of the Eiger Receptors in Circulating Hemocytes Resulted in the Inhibition of TotF Induction in the FB and Enhancement of LG Tumor Growth in mxc^mbn1^

To address the mechanism by which the JNK pathway is activated in hemocytes, we investigated whether Eiger receptors on hemocytes are required for *TotF* induction in the FB of mutant larvae. Eiger binds to two types of receptors encoded by *wgn* and *grnd*, which commonly activate the JNK pathway to induce *upd3* transcription. We quantified the *TotF* mRNA levels in *mxc^mbn1^* larvae harboring hemocyte-specific depletion of *wgn* (*mxc^mbn1^*/*Y*; *He* > *wgnRNAi*) or *grnd* (*mxc^mbn1^*/*Y*; *He* > *grndRNAi*). We performed qRT-PCR using total RNA prepared from normal, *mxc^mbn1^* and mutant larvae harboring these depletions. The hemocyte-specific depletion of *wgn* and *grnd* in *mxc^mbn1^* resulted in 17.1% and 20.0% reductions in mRNA levels compared to those in *mxc^mbn1^* larvae, respectively. These differences were significant (Figure 7).

We quantified the LG size in the normal control (*w*/*Y*), *mxc^mbn1^* larvae (*mxc^mbn1^*/*Y*; *He* > *GFPRNAi*), and those harboring depletion of *wgn* or *grnd* genes encoding one of the Eiger receptors (Appendix A). The average LG size of normal and *mxc^mbn1^* larvae was 0.04 mm^2^ (*n* = 20) and 0.39 mm^2^ (*n* = 20), respectively. In contrast, the average LG sizes of *mxc^mbn1^* larvae harboring the hemocyte-specific depletion of *wgn* and those harboring the *grnd* depletion were 0.51 mm^2^ (*n* = 19) and 0.45 mm^2^ (*n* = 42), respectively. Their LG sizes were 1.3 and 1.1 times larger than those of *mxc^mbn1^* without RNAi, respectively. These differences were significant (Appendix A).

### 2.6. TotB and F Proteins Induced in the FB Were Incorporated into Transplanted Normal Hemocytes in mxc^mbn1^ Larvae but Not in Those in Control Larvae

Tot proteins produced in the FB have an antitumor effect on LG tumors in *mxc^mbn1^* larvae [13]. Thus, we next investigated the mechanisms by which Tot proteins exert these tumor-specific effects. TotB and TotF proteins are incorporated into circulating hemocytes in mutant larvae. To exclude the possibility that these phenotypes may reflect the characteristics of mutant tumor cells, we confirmed whether normal hemocytes transplanted from control larvae contained these Tot proteins. We transplanted normal circulating hemocytes (1.2 × 10^4^ hemocytes on average) expressing GFP from control larvae (*w*/*Y*; *He* > *GFP*) into mutant larvae in which HA-tagged TotB (*mxc^mbn1^*/*Y*; *r4* > *TotB-HA*) or TotF (*mxc^mbn1^*/*Y*; *r4* > *TotF-HA*) were expressed in the FB. Fifteen hours after transplantation, we performed anti-HA immunostaining of circulating hemocytes in *mxc^mbn1^* larvae. We observed a distinctive immunostaining signal for TotB in the cytoplasm of 40.4% of GFP+ hemocytes (*w*/*Y*; *He* > *GFP*) in *mxc^mbn1^* larvae (*n* = 9 out of 23 cells) (Figure 8b′′,e), whereas we found fewer hemocytes exhibiting a TotB signal, the intensity of which was slightly above the background level (9.8% of GFP+ hemocytes) in control larvae (*w*/*Y*; *r4* > *TotB-3HA*) (*n* = 39 cells) (Figure 8a′′,e). This difference was statistically significant (*p* < 0.05, Figure 8e). Consistently, 56.7% of circulating hemocytes in *mxc^mbn1^* larvae showed a distinctive anti-HA immunostaining signal for TotF (*n* = 9 out of 16 cells) (Figure 8d′′). This difference was statistically significant (Figure 8e).

### 2.7. Normal Hemocytes Containing TotF Were Closely Associated with the LGs in mxc^mbn1^ Larvae but Not in Normal Larvae

To confirm the hypothesis that hemocytes incorporating Tot proteins secreted from the FB would be recruited and release antitumor proteins toward the LG tumor, we investigated whether TotF was incorporated into transplanted normal hemocytes and associated with the LG tumor in *mxc^mbn1^* larvae. We transplanted normal hemocytes (1.4 × 10^4^ cells of *w*/*Y*; *He* > *GFP*) in mutant larvae expressing the HA-tagged TotF into their FB *(mxc^mbn1^*/*Y*; *r4* > *TotF-HA*). Fifteen hours after transplantation, we collected the mutant LGs and examined whether the hemocytes closely associated with LG tumors contained Tots via anti-HA immunostaining. Among transplanted normal hemocytes, 22.2% of the GFP+ hemocytes contained Tot proteins in the cytoplasm (Figure 9b′′′,c′′′). In contrast, we did not find any hemocytes containing Tot proteins on the LGs in the control larvae after transplanting the same hemolymph volume from the control larvae (Figure 9a′′′).

## 3. Discussion

In *mxc^mbn1^* mutant larvae, the hemocyte precursor cells in the LG exhibit malignant tumor phenotypes. In response to the tumor, the JAK/STAT pathway is activated in the FB, thereby inducing Tot family proteins with antitumor activity. However, the mechanism by which this induction occurs remained unclear prior to this study. Based on the results mentioned above, we propose the following model: when the Eiger/TNF cytokine secreted from LG tumors binds to the receptors on hemocytes, the JNK pathway in the cell is activated, which induces the expression of Upd3/IL-6 specifically in hemocytes. The hemocytes that recognize the tumor cells by the binding of Eiger are recruited to the FB while secreting Upd3/IL-6. When the Upd3 released from the hemocytes binds to its receptor (Dome) on the FB cells, the JAK/STAT signaling pathway in the cell is activated, and the products of the target genes, TotB and TotF proteins, are induced. These anti-tumor proteins are incorporated into the circulating hemocytes and conveyed toward the LG tumor to induce apoptosis in the tumor cells.

### 3.1. Signal Transfer from the LG Tumor Toward the FB via Circulating Hemocytes Expressing Upd3

In *mxc^mbn1^* larvae carrying the LG tumor, the JAK/STAT pathway in the FB is activated to induce expression of tumor suppressor *Tot* genes [13]. However, to activate the pathway in FBs far away from the tumor, information relevant to the tumor must be conveyed to the FB in which antitumor proteins are produced. Transplanted normal hemocytes are efficiently recruited to LG tumors in *mxc^mbn1^* larvae [9]. This is necessary for suppressing the tumor. Moreover, the present study showed that many mutant and normal hemocytes were recruited to the FB of *mxc^mbn1^* larvae. The binding of Upd3 to its receptor activates the JAK/STAT signaling pathway [11]. In this way, the information is transmitted to the FB and the JAK/STAT pathway is subsequently activated in the FB. Consistently, Upd3 is ectopically expressed in larvae bearing malignant tumors induced by the loss of cell polarity [26]. These data suggest that this cytokine is commonly involved in activating the innate immune pathway in FB regardless of the type of tumor.

We confirmed that the ectopic expression of Upd3 in normal hemocytes transplanted into *mxc^mbn1^* larvae is required for *TotF* induction in the FB. Hemocytes may be recruited to the FB while secreting Upd3 cytokines through the hemolymph to the receptor on the FB’s surface. This activates the JAK/STAT pathway and promotes *TotF* expression. However, we were not able to directly observe the migration of hemocytes on LG tumors towards the FB via live cell imaging. This is a limitation of this study and an issue for future research. Circulating hemocytes also accumulate on imaginal disc tumors in *Drosophila dlg* mutant larvae, and the Toll-mediated innate immune pathway is activated in the FB [8]. This previous finding is consistent with our model that Upd3 in hemocytes is required to signal the FB in LG tumor-bearing larvae. In mammals, the pro-inflammatory cytokine IL-6 is also secreted primarily by lymphocytes and macrophages, which is consistent with findings in *Drosophila* [27]. IL-6 exerts a tumor-suppression effect on cancer cells derived from mammalian breast cancer [28]. These findings support the hypothesis that macrophages are involved in cancer-related signal transfer via cytokines, the expression of which is induced upon tumor recognition—regardless of the species.

### 3.2. Macrophage-like Plasmatocytes May Recognize the Eiger Cytokine from LG Tumors and Convey Information on Tumor Cells to the FB

Our current data suggest that Upd3, which is ectopically induced and secreted from hemocytes, promotes the activation of the JAK/STAT pathway in the FB, whereas low Upd3 expression was observed in the normal hemocytes of tumor-free larvae. Consistently, high Upd3 expression was reported in the hemocytes of mutants with imaginal disc tumors [29]. The Upd3 induced in hemocytes may be used as a signal to notify the FB cells of the presence of tumor cells.

We next investigated how hemocytes could recognize LG tumors. Eiger, a member of the TNF family in *Drosophila*, is highly expressed in LG tumors. Furthermore, its receptor and downstream JNK signaling factors in hemocytes are required for *TotF* expression in the FB. Eiger binds to its receptors, which are composed of Wgn and Grnd, and activates the JNK pathway [15]. Downstream of the JNK pathway, one of the targets, Upd3, is produced [16]. The recognition of LG tumor cells expressing Eiger via receptors on the surface of hemocytes may promote Upd3 expression. Subsequently, these hemocytes are recruited to the FB while expressing Upd3, which induces *TotF* expression in the FB. JNK is activated in the hemocytes of *mxc^mbn1^* larvae [9,24]. Our results confirm that ectopic Upd3 expression and activation of the downstream JNK pathway in hemocytes are required to induce *TotF* expression in the FB. Based on these results, it is inferred that plasmatocytes that accept Eiger via its receptors activate the intracellular JNK pathway. This may trigger the Upd3 induction, and eventually TotF, in FB.

In cultured cells derived from mammalian breast cancer, TNF-α is a potent activator of immune cells via the induction of inflammatory cytokines such as IP-10 [30]. Increased IP-10 levels are associated with the pathology of various inflammatory disorders, including cancer. Therefore, a similar regulatory mechanism may be conserved between *Drosophila* and mammals, in which a TNF ortholog acts as a cytokine that mediates communication between tissues. The mechanism by which hemocytes are recruited to LG tumors also remains unclear; the induction of Eiger may activate the JNK pathway, which in turn leads to cell death [31]. The activation of JNK and induction of its target, Mmp1, have already been reported in LG tumors of *mxc* mutant larvae, resulting in the disassembly of the basement membrane of LG cells [9,24]. The resulting cellular fragments may be phagocytosed by macrophage-like plasmatocytes and recognized as a sign of tissue damage. The chemokine(s) that recruits plasmatocytes to LG tumors needs to be clarified in the future. The expected findings from studies using the *Drosophila* model are also useful for mammalian research on the mechanism of innate immune cells’ recognition of cancer cells and of how they send information to the acquired immune system.

### 3.3. Possible Role of Circulating Hemocytes as a Vector to Transport Secreted Antitumor Proteins, Tots, to the LG Tumor

Many circulating hemocytes, as well as hemocytes recruited to the LG tumor, internalize TotF secreted from the FB. This suggests that hemocytes act as a vector that conveys the antitumor proteins selectively to the tumor. TotF and TotB are incorporated into the cytoplasm of circulating hemocytes and immunostaining signals as Tots are present in small cytoplasmic vesicles [13]. Other AMPs, such as Drosomycin and Defensin, also exhibit antitumor activity by being incorporated into hemocytes and recruited to the LG tumor [7]. Defensin family proteins are conserved in mammals [32], albeit no detailed research on their anticancer effects exists. Neither *Drosophila* AMPs nor Tots, which have anticancer effects, induce apoptosis in normal tissues [7,13]. Therefore, if mammalian AMP-like proteins also have anticancer effects without influence on normal tissues, they would be useful as anticancer drugs with no side effects.

Some mammalian macrophages secrete cytokines such as globule-EGF factor 8 and IL-6, which leads to the tumorigenicity of cancer stem cells [33]. Other mammalian cytokines are also consistently incorporated into the cytoplasm in vesicle-like forms [34]. Similarly, Eiger induces cell death via Rac1-dependent endocytosis [35]. Based on these previous findings, we speculate that *Drosophila* hemocytes may also take up Tot proteins, probably via endocytosis. Subsequently, hemocytes are recruited to the tumor, where they release Tots, probably via exocytosis, as observed in cytotoxic granule discharge from natural killer cells and T lymphocytes, for example [36]. In the near future, it will be necessary to identify the factors involved in Tot dynamics. Additionally, the mechanisms underlying the uptake and release of Tot antitumor proteins need to be clarified.

In mammals, tumor-associated macrophages (TAMs) facilitate disease progression by promoting tumor growth and suppressing adaptive immune responses [37]. TAMs are highly heterogeneous, and they have either a supportive or a suppressive role in anti-tumor immunity [38]. More medical studies have reported that they enhance tumor growth by secreting tumor-promoting cytokines and promoting immunosuppression [37]. Similarly, *Drosophila* hemocytes are associated with imaginal disc tumors and suppress tumor growth promotions [39]. In contrast, macrophages involved in transducing suppressive signals for tumor progression may also be contained among the TAMs, as proposed in this study and others [37,40,41]. Although interorgan communication via immune cells may be a unique concept, immune cell trafficking between different tissues functions to prevent wide-ranging infections [42]. It is also possible to speculate that immune cells that recognize cancer cells in some tissues move towards other tissues to relay the information in order to prevent cancer cells from spreading throughout the body. Our current findings in *Drosophila* provide valuable information to clarify this hypothesis in future cancer biology research.

## 4. Materials and Methods

### 4.1. Drosophila Stocks

*w^1118^*, abbreviated to *w*, was used as a normal control stock. The recessive lethal allele of *mxc*, *mxc^mbn1^* (#6360, Bloomington Drosophila Stock Center (BDSC), Indiana University, Bloomington, IN, USA), was used [7,22]. As *mxc* is an X-linked gene, the mutant male (*mxc^mbn1^*/*Y*) and control male (*w*/*Y*) were used for experiments. For the depletion of JNK factors and several other proteins, we used the following *UAS-RNAi* stocks: *P{w + mC = UAS-GFP.dsRNA.R}142* (*UAS-GFPRNAi*) (BDSC, #BL9330) [13] as a control for dsRNA expression, *P{KK110348}VIE-260B* (*UAS-upd3RNAi*) (Vienna Drosophila Resource Center (VDRC), Vienna, Austria, #v106869) [43], *P{y + t7.7 v + t1.8 = TRiP.HMC03962}attP40* (*UAS-wgnRNAi*) (BDSC, #BL55275) [44], *P{KK109939}VIE-260B* (*UAS-grndRNAi*) (VDRC, #104538) [45], *P{TRiP.HMC03539}attP2* (*UAS-bskRNAi*) (#53310) [46], and *P{y + t7.7 v + t1.8 = TRiP.GL00089}attP2* (*UAS-hepRNAi*) (BDSC, #BL35210) [47]. *P{w[+mC] = upd3-lacZ.Z}5F* (*upd3-LacZ*) (BDSC, #BL-98418) was used to monitor *upd3* gene expression. To induce expression of HA-tagged TotB and TotF, *M{UAS-TotB.ORF.3xHA.GW}ZH-86Fb* (FlyORF, Zurich, Switzerland, #002780) and *M{UAS-TotF.ORF.3 xHA.GW}ZH-86Fb* (FlyORF, #00351) were used [13]. The following GAL4-driver stocks were used: *P{w + mC = He-Gal4.Z}*(BDSC, #BL8699), to induce gene expression in circulating hemocytes [23]; *P{upd3-GAL4}* (provided by N. Perrimon, Harvard Medical School, Boston, MA, USA), to induce gene expression in the medulla zone of the LG [10]; and *P{w + mC = r4-GAL4}3*(BDSC, #BL33832), for FB-specific gene expression [7,13]. All *Drosophila* stocks were maintained on standard cornmeal food at 25 °C as previously described [7,48]. To efficiently induce GAL4-dependent gene expression, individuals carrying the *GAL4 driver* gene and *UAS transgenes* were raised at 28 °C. Fly food was prepared according to a previous procedure [48].

### 4.2. Sample Preparation and Immunostaining of LGs in Larvae

As *mxc^mbn1^* was maintained in stock balanced by *FM7a, P{w[ + mC] = sChFP}1* carrying the marker gene RFP and mature larvae hemizygous for *mxc^mbn1^* at the third-instar stage were selected based on the absence of RFP fluorescence. Normal control males (*w*/*Y*) pupated at 6 d (28 °C) and 7 d (25 °C) after egg laying (AEL), whereas *mxc^mbn1^* males remained in third instar larval stage at 8 d (28 °C) and 10 d (25 °C) AEL. A comparison between the control and *mxc^mbn1^* larvae was performed on the same day that the wandering larvae at the third instar stage appeared to minimize a delay that might allow hyperplastic tissue to grow. To obtain the larvae, five pairs of flies were maintained in culture tubes and allowed to lay eggs for 24 h on food. To compare the sizes of the LGs, a pair of anterior lobes of the LG from mature larvae at the third instar stage were collected and fixed in paraformaldehyde. After staining with DAPI solution (#5748, FUJIFILM Wako Pure Chemical, Osaka, Japan), the fixed LG samples embedded in Vector Shield (#H-1000, Vector Laboratories, Newark, CA, USA) were gently flattened under a covered glass to prepare specimens of a constant thickness, as described in [7]. Specimens were observed under a fluorescence microscope (I×81; Olympus Co., Tokyo, Japan) equipped with a CCD camera (ORCA-R2; Hamamatsu Photonics, Hamamatsu, Japan). MetaMorph (version 7.8 13.0, Molecular Devices Inc., San Jose, CA, USA) was used for image processing. The lobe areas of all LGs were measured from the fluorescence images of DAPI-stained samples using ImageJ (version 1.47, National Institutes of Health, Bethesda, MD, USA).

After the fixed LGs were blocked in 10% NGS in PBST (PBS supplemented with 0.1% Triton X-100), the specimens were incubated with anti-Eiger antibody [25] (dilution 1:500, a gift from M. Miura, Tokyo University) overnight at 4 °C. Alexa Fluor 488-conjugated anti-rabbit and anti-mouse IgG antibodies (1:400, #A11008, and #A1100, Thermo Fisher Scientific, Hillsboro, OR, USA) were used to detect the primary antibody. After DAPI staining of the samples, fluorescence images of the immunostained samples were acquired as described above.

### 4.3. Preparation of Hemocytes in Drosophila Larval Hemolymph and Immunostaining of the Hemocytes

Mature third instar larvae were dissected in a Drosophila Ringer solution (3 mM CaCl_2_-2H_2_O, 182 mM KCl, 46 mM NaCl, 10 mM Tris-base) containing 0.02 µg/mL 1-phenyl-2-thioureic acid (#166-13702, FUJIFILM Wako Pure Chemical, Osaka, Japan). The cells in the hemolymph were allowed to adhere to a glass slide and fixed in 4% paraformaldehyde solution. After staining with DAPI, the specimens were observed under a fluorescence microscope controlled by MetaMorph (version 7.8 13.0; Molecular Devices Inc., San Jose, CA, USA), as described above.

Hemocytes collected and fixed as described above were blocked with 10% NGS in PBS and incubated with anti-JNK/SAPK antibody (1:200, #559304, Merck, Darmstadt, Germany) to detect phosphorylated JNK expression, with anti-β-galactosidase antibody (1:2000, #02150039, MP Biomedicals, Irvine, CA, USA) to monitor the LacZ reporter expression, or with anti-Mmp1 antibody (1:300, 3A6B4, 3B8D12 and 5H7B11, DSHB, Iowa city, IA, USA) [49] overnight at 4 °C. To detect whether HA-TotF was incorporated in the hemocytes localized on the LGs, anti-HA antibody (1:200, #2367, Cell Signaling Technology, Danvers, MA, USA) was used. To detect each primary antibody that reacted with the samples, Alexa Fluor 488- or 555-conjugated anti-rabbit or anti-mouse IgG antibodies (1:400, #A11008, Thermo Fisher, Hillsboro, OR, USA) were used according to the animal species in which the primary antibodies were created. Fluorescence images of the samples were acquired as previously described.

To detect whether HA-TotF protein was incorporated in the hemocytes closely associated with the LGs, the tissues were collected and rinsed with PBS three times for 10 min each before fixation. The LGs, together with associated hemocytes, were fixed with 3.7% paraformaldehyde for 15 min and blocked with 10% NGS in PBS. Anti-HA immunostaining was performed as described above.

### 4.4. Transplantation of Hemocytes in Drosophila Larvae

A constant volume (0.5 μL) of larval hemolymph was injected into a recipient third instar larva using glass needles, as previously described [9]. The needles were prepared from G1.2 capillaries (outer diameter of 1 mm, Narishige Co., Tokyo, Japan) using a grass puller (PN-31, Narishige Co., Tokyo, Japan) and was ground against the side of the microscope glass so that the tip was sharpened to a slant longer than the inner diameter (0.75 mm). This procedure was performed to minimize damage when inserting the needle into the larvae and to prevent the transplanted cells from clogging the tip. The hemolymph was injected within 5 m of dissection of the recipient larvae to avoid melanization and clogging. After injection, the larvae were placed on wet blocking papers for 1 h to recover from the damage and raised on standard food for 15 h before observation.

### 4.5. Quantitative Real-Time PCR (qRT-PCR)

Total RNA was extracted from larval FB at the third-instar stage using TRIzol reagent (Invitrogen, Waltham, MA, USA). The purity of RNA was checked by confirming that the A260 and A280 ratio of each RNA sample was between 1.8 and 2.0. cDNA was synthesized from total RNA using a PrimeScript High-Fidelity RT-PCR Kit (TaKaRa, Clontech Laboratories, Kusatsu, Japan) with oligo dT primers. Real-time PCR was performed using the following qPCR primers: RP49-Fw, 5′-TTCCTGGTGCACAACGTG-3′ and RP49-Rv, 5′-TCTCCTTGCGCTTCTTGG-3′; TotF-Fw, 5′- AGGCACGTCAAATGCTCGC-3′ and TotF-Rv, 5′-TGTTGGTTGTTGTGTGCCCG-3′. The PCR reaction was carried out using a cycling program consisting of initial denaturation at 95 °C for 5 m, followed by 40 cycles at 95 °C for 5 s and 60 °C for 30 s. The temperature was increased from 60 °C to 95 °C at a rate of 0.1 °C/s. Real-time PCR was performed using a Thermal Cycler Dice^®^ Real-Time System III (TaKaRa bio., Kusatsu, Japan) using TB Green Premix Ex Taq II (#RR820A, TaKaRa Bio, Kusatsu, Japan). Each sample was analyzed in triplicate on a PCR plate, and the final results were obtained by averaging three biological replicates. For quantification, the ∆∆Ct method was used to determine the differences between target gene expression and that of the reference gene, Rp49.

### 4.6. Statistical Analysis

Scatter plots were created using GraphPad Prism 9 (GraphPad Software, San Diego, CA, USA) or Microsoft Office Excel 2016 (Microsoft, Redmond, WA, USA) to determine the number of hemocytes in the LGs and FB. The fluorescence intensity of LGs or circulating hemocytes was quantified using ImageJ (Ver.1.54h, National Institutes of Health, Bethesda, MD, USA). Each dataset was assessed using Welch’s *t*-test as previously described [8,23,47]. Before then, an F-test was performed to determine equal or unequal variances. Welch’s *t*-test was performed when the value was less than 0.05 (unequal variance), whereas a Student’s *t*-test was performed when the value was greater than 0.05 (with equal variance). One-way analysis of variance (ANOVA) was used to analyze differences among groups. A *p*-value of 0.05 or less was considered statistically significant.

## Figures and Tables

**Figure 1 ijms-25-13110-f001:**
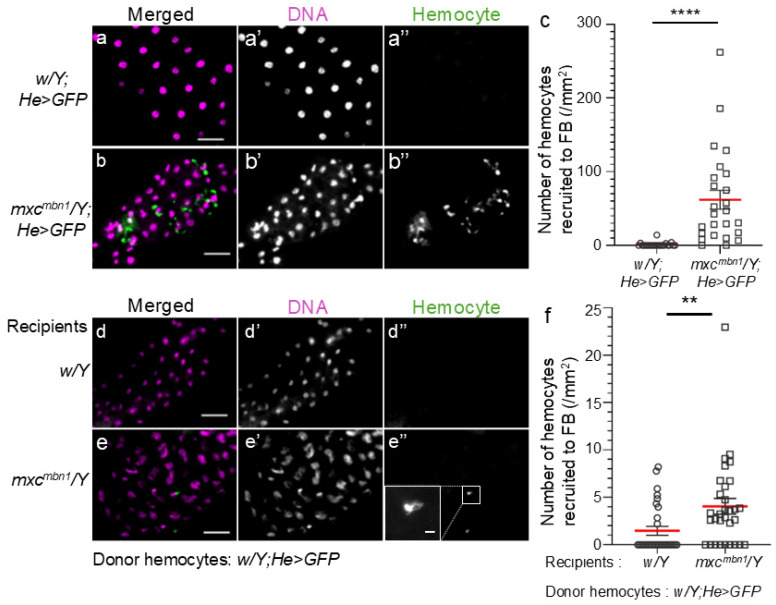
Hemocytes localized on the fat body (FB) in control and *mxc^mbn1^* mutant larvae and normal hemocytes transplanted from control larvae on the FB in control and *mxc^mbn1^* larvae. (**a**,**b**) Fluorescence images of circulating hemocytes labeled by GFP on the DAPI-stained FB in normal control (*w*/*Y*; *He* > *GFP*) (**a**) and *mxc^mbn1^* (*mxc^mbn1^*/*Y*; *He* > *GFP*) (**b**) larvae. The circulating hemocytes are visualized in green in (**a**,**b**) (white in (**a**′′,**b**′′)). DNA is stained in magenta in (**a**,**b**) (white in (**a**′,**b**′)). Scale bars: 100 μm. (**c**) Quantification of the number of hemocytes localized on the FB. The average number of hemocytes per unit area (mm^2^) of FB was calculated (*n* = 22 FB) (15 larvae) (*w*/*Y*; *He* > *GFP*) *n* = 25 (17) (*mxc^mbn1^*/*Y*; *He* > *GFP*). This difference is statistically significant (Welch’s *t*-test, **** *p* < 0.0001). The red line indicates the mean value; error bars indicate the standard error of the mean (SEM). (**d**,**e**) Fluorescence images of normal circulating hemocytes labeled by GFP (*w*; *He* > *GFP*), which were transplanted from control larvae on the DAPI-stained FB in (**d**) normal control (*w*/*Y*) and (**e**) *mxc^mbn1^* mutant (*mxc^mbn1^*/*Y*) larvae. The transplanted normal hemocytes are visualized in green in (**d**,**e**) (white in (**d**′′,**e**′′)). DNA is stained in magenta in d and e (white in (**d**′,**e**′)). Scale bars: 100 μm. (Inset in (**e**′′)). A magnified (3.7 times) image of the FB-associated hemocyte is presented in the inset of (**e**′′). Scale bar in the inset of (**e**′′) represents 10 μm. (**f**) Quantification of the number of hemocytes localized on the FB. The average number of hemocytes per unit area of FB was calculated (*n* = 31 (19)) (*w*/*Y*; *He* > *GFP*, *n* = 31 (23) (*mxc^mbn1^*/*Y*; *He* > *GFP*). This difference is statistically significant (Welch’s *t*-test, ** *p* < 0.01). The red line indicates the mean value; error bars indicate the SEM.

**Figure 2 ijms-25-13110-f002:**
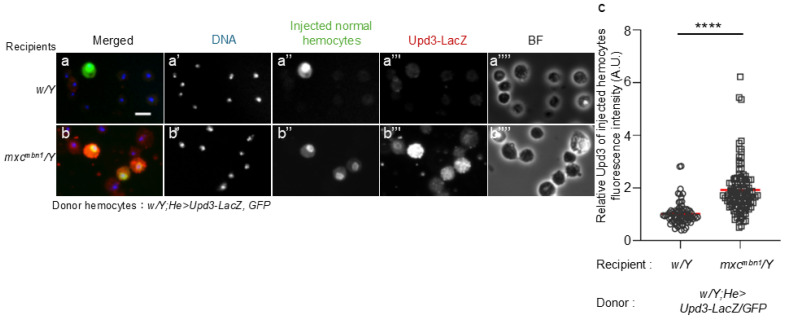
Ectopic expression of the *upd3* gene in normal circulating hemocytes transplanted into control and *mxc^mbn1^* larvae determined using the *upd3*-*LacZ* reporter. (**a**,**b**) Anti-β-gal immunostaining of the circulating hemocytes transplanted from normal larvae at the mature third-instar stage in normal (*w*/*Y*) (**a**) and *mxc^mbn1^* (**b**) larvae (*mxc^mbn1^*/*Y*). Scale bar: 10 µm; blue: DNA (white in (**a**′,**b**′)); green: transplanted hemocytes labeled by GFP (white in (**a**′′,**b**′′)) (*w*/*Y*; *He* > *GFP*/*upd3-LacZ*); red: anti-β-gal immunostaining to monitor the *upd3* expression (white in (**a**′′′,**b**′′′)). BF: brightfield microscopy image in (**a**′′′′,**b**′′′′). (**c**) The relative fluorescence intensity of anti-β-gal immunostaining. The fluorescence intensity of each transplanted hemocyte was quantified and is displayed on the y-axis relative to the fluorescence intensity of the normal control set at 1. X-axis from left to right: normal control (*w*/*Y*) (*n* = 73 transplanted hemocytes (17 larvae)), *mxc^mbn1^* (*mxc^mbn1^*/*Y*) (*n* = 108 (23)). The average fluorescence intensity is shown as a red line. This difference is statistically significant (Welch’s *t* test, **** *p* < 0.0001). Error bars indicate standard error of the mean (SEM).

**Figure 3 ijms-25-13110-f003:**
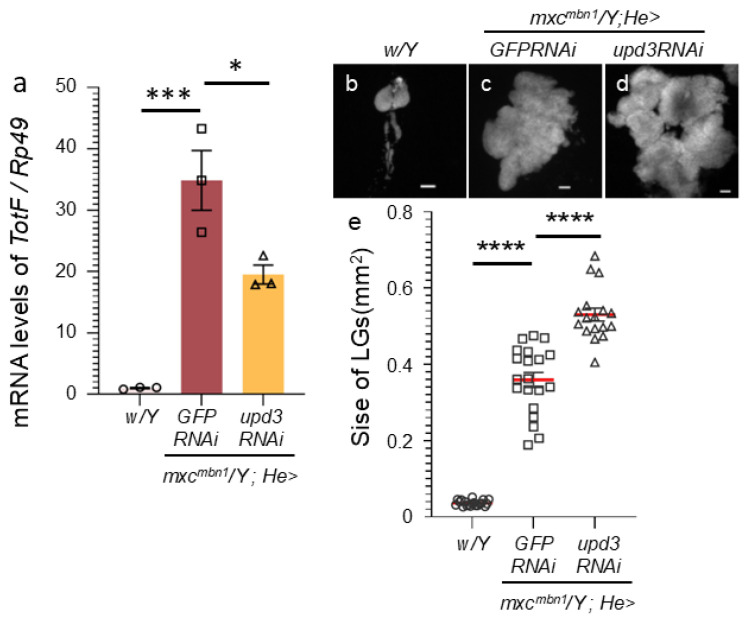
Reduction in mRNA levels of *TotF* in the circulating hemocytes of *mxc^mbn1^* harboring of *upd3* depletion and enhancement of the LG tumor’s growth in the *mxc^mbn1^* larvae harboring *upd3*-depleted circulating hemocytes. (**a**) Quantification of mRNA levels of *TotF* in the FB of mature third instar larvae via qRT-PCR. X-axis from left to right: normal control (*w*/*Y*), hemocyte-specific expression of dsRNA against GFP mRNA in *mxc^mbn1^* mutant larvae (*mxc^mbn1^*/*Y*; *He* > *GFPRNAi*), *mxc^mbn1^* with hemocyte-specific depletion of *upd3* (*mxc^mbn1^*/*Y*; *He* > *upd3RNAi*). The y-axis shows mRNA levels of the target gene (*TotF*) relative to the endogenous control gene (*Rp49*). This difference is statistically significant (one-way ANOVA with Bonferroni correction; * *p* < 0.05, *** *p* < 0.001, *n* = 3). Bars indicate mean mRNA levels of the target gene (mean of three quantification data) and error bars indicate the SEM. (**b**–**d**) DAPI-stained LGs from mature larvae at the late third-instar stage. Normal control (*w*/*Y*) (**b**), *mxc^mbn1^* larvae expressing hemocyte-specific dsRNA against GFP mRNA (*mxc^mbn1^*/*Y*; *He* > *GFPRNAi*) (**c**), and *mxc^mbn1^* larvae harboring hemocyte-specific depletion of *upd3* (*mxc^mbn1^*/*Y*; *He* > *upd3RNAi*) (**d**). Scale bar: 100 µm. (**e**) Quantification of the LG sizes. From left to right: control (*w*/*Y*), *mxc^mbn1^* (*mxc^mbn1^*/*Y*; *He* > *GFPRNAi*), and *mxc^mbn1^* with hemocyte-specific *upd3* depletion (*mxc^mbn1^*/*Y*; *He* > *Upd3RNAi*) larvae. This difference is statistically significant (one-way ANOVA with Bonferroni correction, **** *p* < 0.0001. *n* = 10 pairs of LGs (20 larvae) (*w*/*Y*), *n* = 10 (20) (*mxc^mbn1^*/*Y*; *He* > *GFPRNAi*), *n* = 8 (17) (*mxc^mbn1^*/*Y*; *He* > *upd3RNAi*)). Red lines indicate means and error bars represent the standard errors of the mean (SEM).

**Figure 4 ijms-25-13110-f004:**
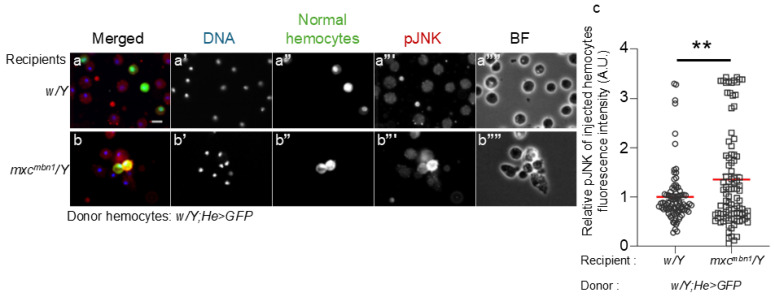
Anti-phosphorylated JNK (pJNK) immunostaining of normal circulating hemocytes transplanted into control and *mxc^mbn1^* larvae to detect activation of the JNK pathway. (**a**,**b**) Fluorescence images of circulating hemocytes transplanted from normal larvae (*w*/*Y*; *He* > *GFP*) in normal (*w*/*Y*) (**a**) and *mxc^mbn1^* larvae (*mxc^mbn1^*/*Y*) (**b**). Scale bar: 10 µm; blue, DNA (white in (**a**′,**b**′)); green, transplanted hemocytes (white in (**a**′′,**b**′′)); red, the hemocytes harboring the activated JNK (pJNK) (white in (**a**′′′,**b**′′′)). BF: brightfield microscopy image in (**a**′′′′,**b**′′′′). (**c**) The relative fluorescence intensity of anti-pJNK immunostaining. The fluorescence intensity of each transplanted hemocyte was quantified and is displayed on the y-axis relative to the fluorescence intensity of the normal control set at 1. X-axis from left to right: normal control (*w*/*Y*) (*n* = 88 transplanted hemocytes (19 larvae)), *mxc^mbn1^* (*mxc^mbn1^*/*Y*) (*n* = 96 (17)) larvae. The average fluorescence intensity is shown as a red line. This difference is significant (Welch’s *t* test, ** *p* < 0.01). Error bars indicate the SEM.

**Figure 5 ijms-25-13110-f005:**
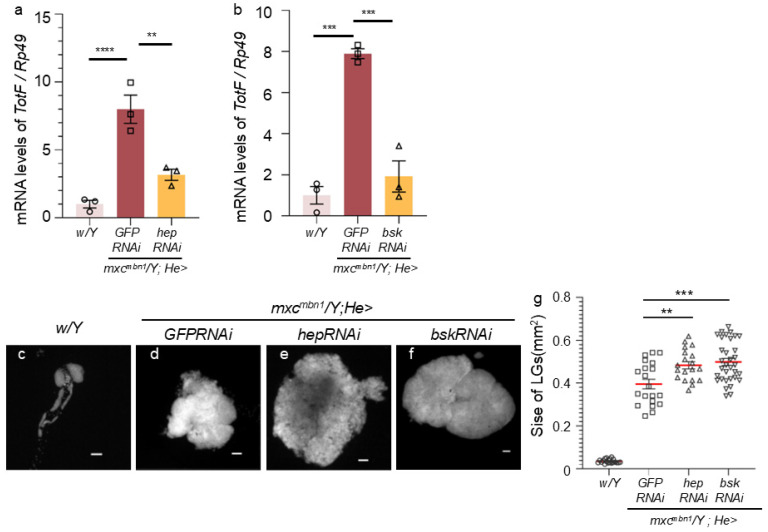
Reduced *TotF* mRNA levels in the FB of *mxc^mbn1^* larvae harboring circulating hemocyte-specific depletion of JNK pathway’s components. JNKK encoded by *hep* and JNK by *bsk*. (**a**,**b**) Quantification of *TotF* mRNA levels in the FB of mature third instar larvae via qRT-PCR. X axis from left to right: normal control (*w*/*Y*), *mxc^mbn1^* expressing control dsRNA against *GFP*dsRNAs specifically in hemocytes (*mxc^mbn1^*/*Y*; *He* > *GFPRNAi*), and hemocyte-specific depletion of *JNKK* (*hep*) in *mxc^mbn1^* (*mxc^mbn1^*/*Y*; *He* > *hepRNAi*) (**a**) or hemocyte-specific depletion of *JNK* (*bsk*) in *mxc^mbn1^* (*mxc^mbn1^*/*Y*; *He* > *bskRNAi*) (**b**). The y-axis shows the mRNA levels of the target gene (*TotF*) relative to the endogenous control gene (*Rp49*). This difference is statistically significant (one-way ANOVA with Bonferroni correction, ** *p* < 0.01, *** *p* < 0.001, **** *p* < 0.0001, *n* = 3. Bars indicate relative mRNA levels of the target gene (mean of three quantification data measurements), and error bars indicate the SEM. (**c**–**f**) DAPI-stained images of LGs from mature third-instar larvae. (**c**) Normal control (*w*/*Y*); (**d**) *mxc^mbn1^* mutant control (*mxc^mbn1^*/*Y*; *He* > *GFPRNAi*); (**e**) *mxc^mbn1^* mutant larvae with hemocyte-specific *hep* depletion (*mxc^mbn1^*/*Y*; *He* > *hepRNAi*); (**f**) *mxc^mbn1^* mutant larvae with hemocyte-specific *bsk* depletion (*mxc^mbn1^*/*Y*; *He* > *bskRNAi*). All scale bars represent 100 μm. (**g**) Quantification of LG sizes. From left to right: normal control (*w*/*Y*) (*n* = 10 pairs of LG (19 larvae)), *mxc^mbn1^* expressing hemocyte-specific dsRNA against *GFP* mRNA (*mxc^mbn1^*/*Y*; *He* > *GFPRNAi*) (*n* = 10 (14)), *mxc^mbn1^* harboring hemocyte-specific *hep* depletion (*mxc^mbn1^*/*Y*; *He* > *hepRNAi*) (*n* = 10 (20)), and those harboring *bsk* depletion (*mxc^mbn1^*/*Y*; *He* > *bskRNAi*) (*n* = 17 (18)). This difference is statistically significant (one-way ANOVA with Bonferroni correction, ** *p* < 0.01, *** *p* < 0.001). Red lines indicate means and error bars indicate the SEM.

**Figure 6 ijms-25-13110-f006:**
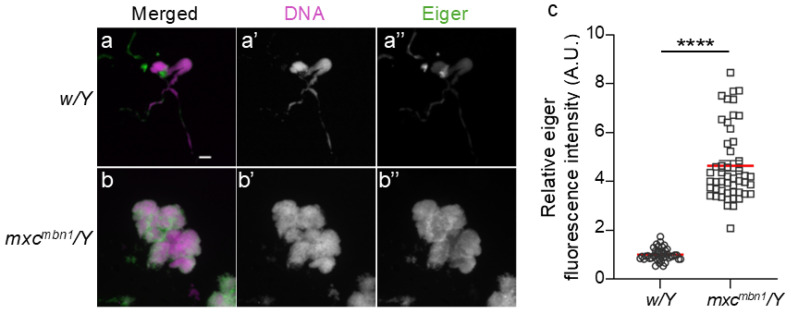
Increased expression of Eiger in the LGs of *mxc^mbn1^* larvae compared to that of normal LGs. (**a**,**b**) Fluorescence images of LGs from mature third instar larvae immunostained with anti-Eiger antibody. a represents normal control (*w*/*Y*) and b represents LG of *mxc^mbn1^* larvae (*mxc^mbn1^*/*Y*). Scale bar is 100 µm; magenta indicates DNA staining (white in (**a**′,**b**′)) and green indicates antibody staining signal (white in (**a**′′,**b**′′)). (**c**) Quantification of fluorescence intensity after anti-Eiger immunostaining of LGs from control and *mxc^mbn1^* larvae. Relative values are shown on the y-axis, with the mean fluorescence intensity of the normal control as 1. From left to right, the X-axis shows the fluorescence intensity in LGs of normal control (*w*/*Y*) (*n* = 21 pairs of LG (35 larvae)) and *mxc^mbn1^* larvae (*mxc^mbn1^*/*Y*) (*n* = 25 (27)). This difference is statistically significant (Welch’s *t*-test, **** *p* < 0.0001). The red line indicates the mean value. Error bars indicate the SEM.

**Figure 7 ijms-25-13110-f007:**
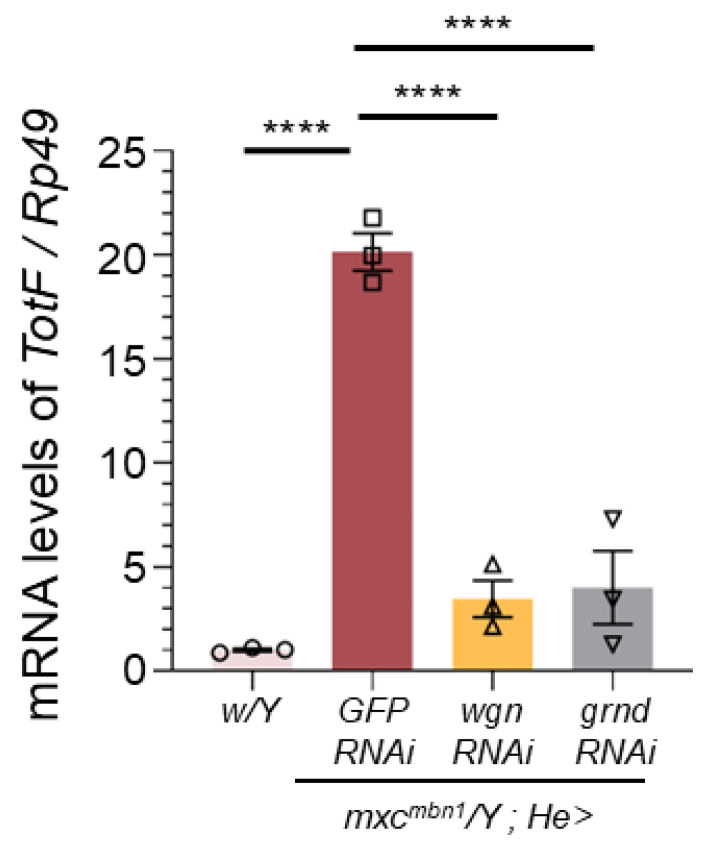
A reduction in *TotF* mRNA levels in circulating hemocytes harboring depletion of *wgn* or *grnd* in *mxc^mbn1^* larvae. Quantification of the *TotF* mRNA levels in the FB of mature third instar larvae with circulating hemocytes harboring depletion of receptor components (Wgn and Grnd) was performed using qRT-PCR. X-axis from left to right: normal control (*w*/*Y*), *mxc^mbn1^* larvae hemocyte-specific expression of dsRNA against *GFP* mRNA (*mxc^mbn1^*/*Y*; *He* > *GFPRNAi*), and *mxc^mbn1^* larvae with hemocyte-specific depletion of *wgn* (*mxc^mbn1^*/*Y*; *He* > *wgnRNAi*), *mxc^mbn1^* with the depletion of *grnd* (*mxc^mbn1^*/*Y*; *He* > *grndRNAi*). The y-axis shows the mRNA levels of *TotF* relative to the endogenous control gene (*Rp49*). Bars indicate relative mRNA levels of the target gene (mean of triplicate quantification data), and error bars indicate the SEM. This difference is statistically significant (**** *p* < 0.0001, one-way ANOVA with Bonferroni correction).

**Figure 8 ijms-25-13110-f008:**
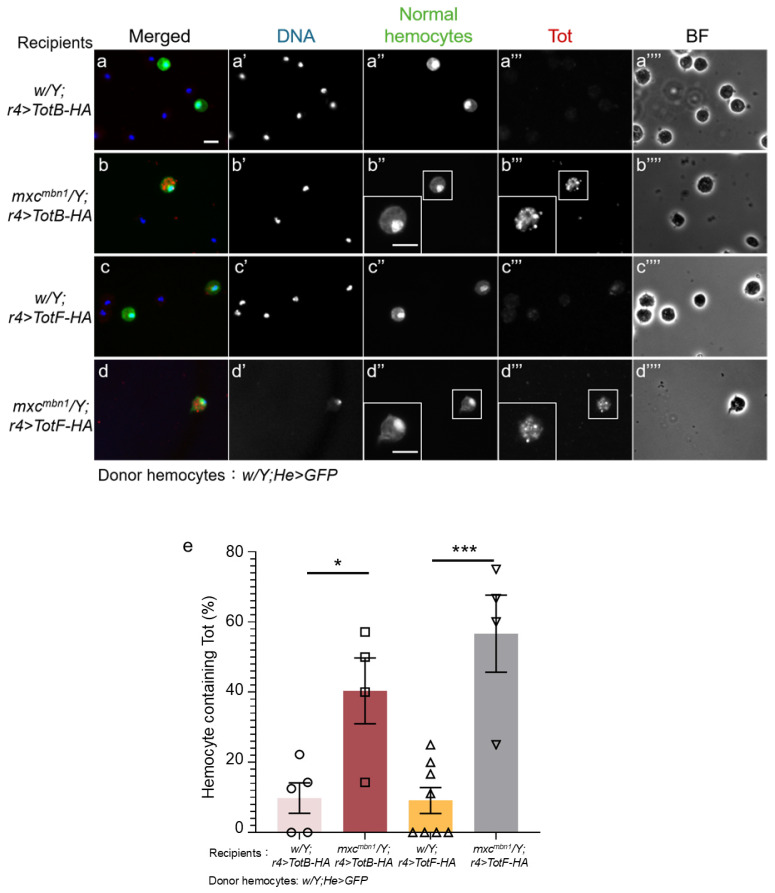
Uptake of TotB and TotF proteins into transplanted normal hemocytes in *mxc^mbn1^* larvae. (**a**–**d**) Anti-HA immunostaining of circulating hemocytes to detect HA-tagged TotB (**a**,**b**) and TotF (**c**,**d**), induced in the FB in normal control larvae (*w*/*Y*; *r4* > *TotB-HA*) (**a**), (*w*/*Y*; *r4* > *TotF-HA*) (**c**), and *mxc^mbn1^* larvae (*mxc^mbn1^*/*Y*; *r4* > *TotB-HA*) (**b**), and (*mxc^mbn1^*/*Y*; *r4* > *TotF-HA*) (**d**). The same hemolymph volume containing circulating hemocytes from control larvae (*w*; *He* > *GFP*) was transplanted into the mutant larvae. The transplanted normal hemocytes labeled by GFP fluorescence are colored in green in a–d (white in (**a**′′–**d**′′)). The anti-HA immunostaining signal of the hemocytes is in red in a–d (white in (**a**′′′–**d**′′′)). DNA is blue in a–d (white in (**a**′–**d**′)). BF: brightfield microscopy image in (**a**′′′′–**d**′′′′). Magnified (1.9 times) images of the hemocytes are presented in the insects in (**b**′′,**b**′′′,**d**′′,**d**′′′). All scale bars represent 10 μm. (**e**) Percentages of normal hemocytes containing TotB or TotF proteins induced in the FB in control and *mxc^mbn1^* mutant larvae. X-axis from left to right: normal control larvae in which HA-tagged TotB was ectopically expressed in the FB of *w*/*Y*; *r4* > *HA-TotB* (*n* = 5 experiments (8 larvae)), *mxc^mbn1^* /*Y*; *r4* > *HA-TotB* (*n* = 4 (6))*, w*/*Y*; *r4* > *HA-TotF* (*n* = 8 (16)), and *mxc^mbn1^* /*Y*; *r4* > *HA-TotF* (*n* = 4 (6)) larvae. This difference is statistically significant (* *p* < 0.05, *** *p* < 0.001, one-way ANOVA with Bonferroni correction). Bars indicate the average frequencies of GFP+ hemocytes containing TotB or TotF, and error bars indicate SEM.

**Figure 9 ijms-25-13110-f009:**
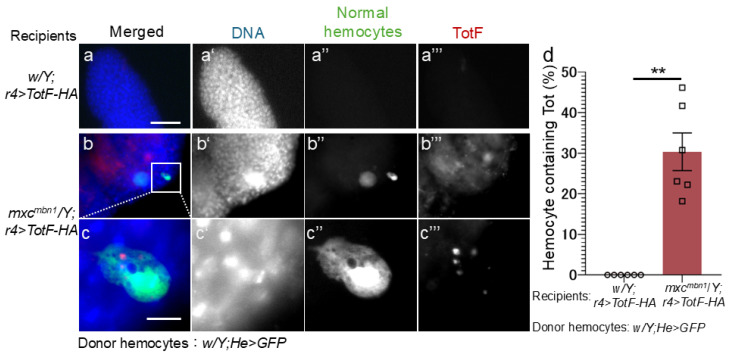
The transplanted normal hemocytes on the LG tumor incorporated the TotF protein that is expressed in the LGs in *mxc^mbn1^* larvae. (**a**–**c**) Anti-HA immunostaining of normal circulating hemocytes expressing GFP transplanted from normal larvae (*w*/*Y*; *He* > *GFP*) on the LG lobes to detect HA-tagged TotF in control (**a**) and *mxc^mbn1^* (**b**,**c**) larvae expressing HA-tagged TotF. The transplanted hemocyte enclosed by a square in (**b**) is magnified (4.8 times) in (**c**). The transplanted hemocytes are in green in a-c (white in (**a**′′–**c**′′)). The anti-HA immunostaining signal (Tot) corresponding the TotF signal is in red in a-c (white in (**a**′′′–**c**′′′)). PC cells possess the ability to incorporate Tot-HA proteins. DNA is colored in blue in a-d (white in (**a**′–**c**′)). The weak blue signal around the hemocyte in c represents DNA staining of the LG cells on which the hemocyte was localized. Scale bars in a and c represent 50 μm and 10 μm, respectively. (**d**) Percentages of transplanted normal hemocytes containing TotF (donor; *w*/*Y*; *He* > *GFP*), which were localized on the LGs in *mxc^mbn1^* larvae (recipient). Recipient larvae on X-axis from left to right: control larvae in which HA-tagged TotF was ectopically expressed in the LGs of *w*/*Y*; *r4* > *TotF-HA* (*n* = 6 experiments (17 pairs of LG from 34 larvae)). (No transplanted hemocytes were found in the LGs in the control recipient larvae and *mxc^mbn1^* /*Y*; *r4* > *TotF-HA* (*n* = 6 (20)). This difference is significant (Welch’s *t* test, ** *p* < 0.01). Error bars indicate the SEM.

## Data Availability

The datasets generated and/or analyzed in the current study are available from the corresponding author upon reasonable request.

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
