# Peer review of "Macrophage-like Blood Cells Are Involved in Inter-Tissue Communication to Activate JAK/STAT Signaling, Inducing Antitumor Turandot Proteins in Drosophila Fat Body via the TNF-JNK Pathway"

_ijms, 2024, doi:10.3390/ijms252313110_

Round 1

Reviewer 1 Report

Comments and Suggestions for Authors

The communication between tissues and blood cells is important and not easy to study. In their submitted manuscript, Juri Kinoshita et al. demonstrated that cells expressing Upd3 are recruited to the FB after Eiger activates the JNK pathway in hemocytes on the tumor. Upd3 then activates JAK/STAT to induce the expression of antitumor proteins, which is intriguing. Building on these findings, the authors illustrate the intricate communication between tissues via blood cells during tumor suppression. To me, this work represents a step forward in the understanding of communication between tissues via blood cells. I have the following questions/comments.

1.     Please re-beautify Figure 1, Figure 2, Figure 4, Figure 8, Figure 9, there are some empty regions in Donor/ recipient, which makes figure look weird.   

2.     Figure 1c and 1f, Please indicate how many flies you used for the statistics in the figure legends.

3.     Figure 2c, 4c, 5g, 6c, Please indicate how many flies you used for the statistics in the figure legends.

4.     Figure 3, Figure 5a, b, Figure 7, please use WB to check the protein level if you can get the toft antibody.

5.     The method section requires substantial improvement to enhance its clarity and reproducibility for other researchers.

6.     Some modifications/corrections in the manuscript may be needed.

Comments on the Quality of English Language

N/A

Author Response

Reviewer 1

Comments and Suggestions for Authors

The communication between tissues and blood cells is important and not easy to study. In their submitted manuscript, Juri Kinoshita et al. demonstrated that cells expressing Upd3 are recruited to the FB after Eiger activates the JNK pathway in hemocytes on the tumor. Upd3 then activates JAK/STAT to induce the expression of antitumor proteins, which is intriguing. Building on these findings, the authors illustrate the intricate communication between tissues via blood cells during tumor suppression. To me, this work represents a step forward in the understanding of communication between tissues via blood cells. I have the following questions/comments.

 First, we appreciate this reviewer for his/her careful reading and for providing thoughtful comments.

1.Please re-beautify Figure 1, Figure 2, Figure 4, Figure 8, Figure 9, there are some empty regions in Donor/ recipient, which makes figure look weird.   

(Response) According to the reviewer’s request, we revised the relevant five figures (lines 144, 178, 237, 368, 397) so that unnecessary empty space would not lead to misunderstanding among readers.

  1. Figure 1c and 1f, Please indicate how many flies you used for the statistics in the figure legends.

(Response)The total number of larvae used for each experiment was added to each Figure legend as requested by the reviewer in lines 151-152 (Figure 1c) and line 161 (Figure 1f).

  1. Figure 2c, 4c, 5g, 6c, Please indicate how many flies you used for the statistics in the figure legends.

(Response)As requested, the number of larvae used was added to each Figure legend in lines 187 (Figure 2c), 245-246 (Figure 4c), 282-284 (Figure 5g), and line 309 (Figure 6c). In addition, we also added the number of larvae for other quantitative figures that the reviewer did not point out: lines 378-379 (Figure 8e) and 412-414 (Figure 9d).

  1. Figure 3, Figure 5a, b, Figure 7, please use WB to check the protein level if you can get the toft antibody.

(Response)

Unfortunately, we have not obtained specific antibodies against the TotF or TotB, although we have tried to generate them. Any antibodies against these two proteins were neither available commercially nor from other laboratories. It may not be easy to make antibodies against these small secreted proteins, including antimicrobial peptides. Thus, we have not performed western blot experiments to verify the results indicating alteration of these mRNAs. We will generate the antibodies and check the protein levels using them in our future study.

  1. The method section requires substantial improvement to enhance its clarity and reproducibility for other researchers.

(Response) According to the reviewer’s request, along with the comment raised by Reviewer 2, we added 12 modifications/corrections describing more details about the experimental methods (545, 587-588, 613-615, 621-625, 632-633), including dilution of antibodies used for immunostaining for each antibody (584, 586, 601, 603, 604, 606, 609) in Materials and Methods.

  1. Some modifications/corrections in the manuscript may be needed.

(Response) According to the reviewer’s suggestion, we carefully read the manuscript repeatedly, and made more than 100 revisions to improve the clarity of the content and improve the English quality in the text. Please refer to the word tract in the Word for details of updates in this version.

Reviewer 2 Report

Comments and Suggestions for Authors   This manuscript, titled "Macrophage-like Blood Cells are involved in Inter-tissue Communication to activate JAK/STAT Signaling that induces Antitumor Turandot Proteins in Drosophila Fat Body via the TNF-JNK pathway," provides valuable insights into tumor suppression mechanisms in Drosophila melanogaster. The study elegantly investigates the interplay between hemocytes, the JAK/STAT signaling pathway, and tumor suppression via antitumor proteins. The findings are significant, with implications for understanding innate immunity and tumor biology. However, some aspects of the manuscript could benefit from further refinement.  My comments are described as followings:   Comments: 1. The discussion effectively places findings in context but could address potential limitations more directly, such as how these findings translate to mammalian systems or other models.  2. In all figures, please check the statistics description. For example, in the legend of figure 1, authors must provide the significant on the quantitative results.   3. Figure legends are generally clear but could benefit from including additional methodological details (e.g., staining conditions or magnification levels)  4. Authors used RT-qPCR as an approach to validate the target genes in this study. The quality of mRNA is critical in this assessment. Authors must provide proof to assess the quality of mRNA used in this paper. 5. I encourage authors to provide the graphic summary. 6. With minor revisions and enhancements to clarity, the manuscript has the potential to make a substantial impact.

Author Response

Reviewer ï¼’

Comments and Suggestions for Authors

  This manuscript, titled "Macrophage-like Blood Cells are involved in Inter-tissue Communication to activate JAK/STAT Signaling that induces Antitumor Turandot Proteins in Drosophila Fat Body via the TNF-JNK pathway," provides valuable insights into tumor suppression mechanisms in Drosophila melanogaster. The study elegantly investigates the interplay between hemocytes, the JAK/STAT signaling pathway, and tumor suppression via antitumor proteins. The findings are significant, with implications for understanding innate immunity and tumor biology.

However, some aspects of the manuscript could benefit from further refinement.  

My comments are described as followings:  

First, we appreciate this reviewer for his/her careful reading and for providing thoughtful comments.

1. The discussion effectively places findings in context but could address potential limitations more directly, such as how these findings translate to mammalian systems or other models. 

(Response)

We appreciate the reviewer’s comment letting us to notice that the claim how the present findings in Drosophila can translate to human and other organisms’ system is insufficient. In accordance with the reviewer’s comment, we revised and added the sentences describing how the current results can be applied to mammalian carcinogenesis and other Drosophila tumor models in the discussion (lines 421-430, 448-451, 454-4594, 494-496, 506-510, and 521-535).

  1. In all figures, please check the statistics description. For example, in the legend of figure 1, authors must provide the significant on the quantitative results. 

(Response)

According to the reviewer’s suggestion, we added a sentence, “The difference between control and experiment is significant.” describing a statistical description in a legend of each quantitative figure, in addition to the p-value and statistical method (lines 152,161, 188, 205, 212, 246, 276, 284, 310, 335, 379-380, 414). We considered that the difference is statistically significant if the p-value is 0.05 or less.

  1. Figure legends are generally clear but could benefit from including additional methodological details (e.g., staining conditions or magnification levels) 

(Response)According to the reviewer’s comment, along with reviewer 1’s request, we added several sentences and phrases describing more detailed methodological information in Materials and Methods, such as the dilution of antibodies used (lines 584-586, 601-609). The magnification rates of the inset images in Fig. 1e”, Fig. 8B”, b”’, Fig. 8d”, d”’ were also added in the relevant figure legends (lines 158, 375 and 403, respectively) .

  1. Authors used RT-qPCR as an approach to validate the target genes in this study. The quality of mRNA is critical in this assessment. Authors must provide proof to assess the quality of mRNA used in this paper.

(Response)

Total RNA used qRT-PCR was extracted from larval FB at the third instar stage using TRIzol reagent. We have selected living larvae at the third instar stage for RNA extraction. Dead or dying larvae or individuals that have developed into the white pupal stage were not selected. The purity of RNA was checked by confirming that an A260 and A280 ratio of each RNA sample is between 1.8–2.0 so that the RNA samples did not contain contaminated proteins or DNA. If the purity of the RNA sample was not sufficiently high, or if the amount of RNA recovered from a certain volume of tissues was low, we did not use the RNA samples for cDNA synthesis and repeated the RNA preparation. We added an underlined sentence in Materials and Methods 4.5. However, we have not examined the integrity of the RNA samples by Agarose gel electrophoresis or other methods.

  1. I encourage authors to provide the graphic summary.

(Response)

We appreciate the reviewer’s encouragement. We submitted the graphic abstract with the manuscript. We hope that the GA could help the understanding of the reviewers and readers.

  1. With minor revisions and enhancements to clarity, the manuscript has the potential to make a substantial impact.

(Response)

According to the reviewer’s suggestion, we carefully read the manuscript repeatedly and made more than 100 corrections/modifications to improve the clarity of the content, and the quality of English in the manuscript following the proof reading by the MDPI author services (English ID: english-87745, (please see the enclosed certificate issued by the MDPI author services)). Please refer to the word tract in the Word for details of updates in this version.

Reviewer 3 Report

Comments and Suggestions for Authors

This study provides valuable insights into the tumor-suppressive mechanisms in Drosophila, emphasizing the intricate interplay between hemocytes and tumor tissues. The identification of the Eiger-JNK-Upd3 axis as a critical pathway in regulating JAK/STAT activation and Tot protein induction highlights the complex crosstalk between immune cells and tumors. By demonstrating how hemocytes mediate tumor-related signals and contribute to antitumor responses, this work advances our understanding of tissue communication in tumorigenesis and sheds light on potential parallels in other organisms. Overall, this paper provides compelling evidence through rigorous Drosophila genetic manipulations and is recommended for publication after minor revisions.

Here are a few of my minor concerns:

Some terms need to be explained in detail, such as "LG tumor."

In Figures 1a-b, how is it determined that the fat body cells are involved? Does DAPI in circulating hemocytes exclusively label the fat body?

I could not locate Figure S1d. It is suggested to include this data alongside Figure 3.

The supplementary figures (Figure S) mentioned in the text are not available.

Author Response

Reviewer 3

This study provides valuable insights into the tumor-suppressive mechanisms in Drosophila, emphasizing the intricate interplay between hemocytes and tumor tissues. The identification of the Eiger-JNK-Upd3 axis as a critical pathway in regulating JAK/STAT activation and Tot protein induction highlights the complex crosstalk between immune cells and tumors. By demonstrating how hemocytes mediate tumor-related signals and contribute to antitumor responses, this work advances our understanding of tissue communication in tumorigenesis and sheds light on potential parallels in other organisms. Overall, this paper provides compelling evidence through rigorous Drosophila genetic manipulations and is recommended for publication after minor revisions.

 Here are a few of my minor concerns:

First, we appreciate this reviewer for his/her careful reading and for providing thoughtful comments.

1. Some terms need to be explained in detail, such as "LG tumor."

(Response)

We added the short explanation in line 51-52. “As a result, apoptosis is observed in the tumors generated in lymph gland (LG) (LG tumors) due to the induction of AMPs and Tots, and tumor cell proliferation is also inhibited in Tots. However, the mechanism by which the effect of Tots on LG tumors located away from the FB has not yet been characterized.”

2. In Figures 1a-b, how is it determined that the fat body cells are involved? Does DAPI in circulating hemocytes exclusively label the fat body?

(Response)

As the reviewer pointed out, DAPI stains all the nuclei of FB and the associated hemocytes on it. Therefore, we transplanted hemocytes expressing GFP into the recipient larvae that did not carry the GFP gene and examined whether GFP fluorescence was observed on the FB of the recipients.

Fig. 1e shows that the transplanted normal hemocytes are also localized to the FBs of the mxc mutant. Furthermore, in the supplementary data uploaded to the web (see the images, which are only for review in a repository), we have confirmed by confocal microscopy that the transplanted normal hemocytes are localized on the surface (but not the inside) of the FB in normal and mutant larvae. Based on these results, we concluded that normal hemocytes also accumulate on the FB of mxc mutant bearing the LG tumor.

3. I could not locate Figure S1d. It is suggested to include this data alongside Figure 3.

(Response)

In response to the reviewer's recommendation, we have revised Figure S1 to include it within Figure 3b-e.

4. The supplementary figures (Figure S) mentioned in the text are not available.

(Response)

Although we might be misinterpreting this comment, we mentioned Figure S1 (previous Figure S2) in the text in lines 235-236. Supplementary figure 2 (previous figure S3) is mentioned in the text in lines 338-345. The description about previous figure S1 (now in Figure 3b-e) is inserted in lines 216-223.

Round 2

Reviewer 1 Report

Comments and Suggestions for Authors

The authors have answered my questions. I don't have other questions.